# The Interplay of Permeability, Metabolism, Transporters, and Dosing in Determining the Dynamics of the Tissue/Plasma Partition Coefficient and Volume of Distribution—A Theoretical Investigation Using Permeability-Limited, Physiologically Based Pharmacokinetic Modeling

**DOI:** 10.3390/ijms242216224

**Published:** 2023-11-12

**Authors:** Lu Gaohua, Mian Zhang, Caroline Sychterz, Ming Chang, Brian James Schmidt

**Affiliations:** Clinical Pharmacology & Pharmacometrics, Bristol Myers Squibb, Lawrenceville, NJ 08540, USA; mian.zhang@bms.com (M.Z.); caroline.sychterz@bms.com (C.S.); ming.chang@bms.com (M.C.); brian.schmidt@bms.com (B.J.S.)

**Keywords:** physiologically based pharmacokinetic (PBPK) model, tissue/plasma partition coefficient (Kp), volume of distribution (Vd), permeability, metabolism, transporter, perfusion-limited, permeability-limited

## Abstract

A permeability-limited physiologically based pharmacokinetic (PBPK) model featuring four subcompartments (corresponding to the intracellular and extracellular water of the tissue, the residual plasma, and blood cells) for each tissue has been developed in MATLAB/SimBiology and applied to various what-if scenario simulations. This model allowed us to explore the complex interplay of passive permeability, metabolism in tissue or residual blood, active uptake or efflux transporters, and different dosing routes (intravenous (IV) or oral (PO)) in determining the dynamics of the tissue/plasma partition coefficient (Kp) and volume of distribution (Vd) within a realistic pseudo-steady state. Based on the modeling exercise, the permeability, metabolism, and transporters demonstrated significant effects on the dynamics of the Kp and Vd for IV bolus administration and PO fast absorption, but these effects were not as pronounced for IV infusion or PO slow absorption. Especially for low-permeability compounds, uptake transporters were found to increase both the Kp and Vd at the pseudo-steady state (Vdss), while efflux transporters had the opposite effect of decreasing the Kp and Vdss. For IV bolus administration and PO fast absorption, increasing tissue metabolism was predicted to elevate the Kp and Vdss, which contrasted with the traditional derivation from the steady-state perfusion-limited PBPK model. Moreover, metabolism in the residual blood had more impact on the Kp and Vdss compared to metabolism in tissue. Due to its ability to offer a more realistic description of tissue dynamics, the permeability-limited PBPK model is expected to gain broader acceptance in describing clinical PK and observed Kp and Vdss, even for certain small molecules like cyclosporine, which are currently treated as perfusion-limited in commercial PBPK platforms.

## 1. Introduction

Physiologically based pharmacokinetic (PBPK) modeling and simulation goes back to the work of Teorell in 1937 [1,2]. It is increasingly used in drug discovery and development by the pharmaceutical industry, academia, and health authorities [3,4,5,6,7]. Its application has been recently further fueled by the acceptance of PBPK results on drug labels [6,8,9]. Under the paradigm of model-informed drug development, major health authorities have provided detailed regulatory guidance on PBPK submissions [10,11,12].

Irrespective of using the commercial ready-to-use PBPK platforms (such as GastroPlus [13] and Simcyp [14]) or developing bespoke PBPK models in general programming software (such as MATLAB and R), the tissue/plasma partition coefficient (Kp) and volume of distribution at steady state (Vdss) are two essential parameters to retrospectively describe or prospectively project systemic and tissue exposures. These two parameters are linked together through the anatomic physiological volumes of blood and tissue [15,16].

The mechanistic prediction of the Kp could find its roots in the pioneering works on the solubility of anesthetic gas in blood and tissue from the early 1960s [17,18]. On the other hand, the mechanistic prediction of Vdss originates from the works of plasma and tissue binding in the 1970s [19,20,21]. Recent developments in this field are represented by the works of Poulin–Theil [22,23] and Rodgers–Rowland [24,25]. Both have been implemented in GastroPlus and Simcyp, and are widely accepted by the scientific community. Of note, these mechanistic methods have their predictions based on tissue binding and passive permeability in the context of defining tissue distribution. However, they rely on several critical assumptions, such as well-stirred tissue at a steady state, the absence of active transporters, or metabolism within the tissue and blood. PBPK models based on these assumptions are commonly termed perfusion-limited, because they assume rapid equilibration between tissue vascular and extravascular spaces, with tissue perfusion considered a limiting factor in the process.

However, these technical assumptions present clear caveats. After a single instantaneous dose administration to the human body, only a pseudo-steady state can be achieved when the redistribution and elimination reach an equilibrium. The human body operates as an open system, with drug-metabolizing enzymes present in the tissue cells or beyond, such as in tissue residual blood (plasma and blood cells) and in the circulating plasma and blood cells. Further complexity is added by various active transporters on the cell membranes to uptake or efflux xenobiotics. Understanding the intricate interplay between metabolic enzymes and active transporters poses a significant challenge in elucidating drug absorption, distribution, metabolism, excretion, and drug–drug interactions [26,27,28,29,30].

The kinetics of tissue concentration in the traditional perfusion-limited PBPK model is often described using a linear ordinary differential equation (ODE), such as
(1)VtissuedCtissuedt=QtissueCblood − CtissueKptissueBP − CLint·CtissueKptissueBP
where Vtissue, Ctissue, Cblood, Qtissue, BP, and CLint are the tissue volume, tissue concentration, blood concentration, tissue blood flow, blood/plasma partition coefficient, and tissue intrinsic clearance, respectively. Kptissue is the tissue/plasma partition coefficient constant that can be predicted mechanistically, as previously mentioned.

The tissue/plasma partition coefficient at the pseudo-steady state (Kpss) can be derived by setting dCtissue/dt = 0 in the above ODE (Equation (1)). Therefore, the Kpss and Vdss at the pseudo-steady state are given as below:(2)$${{Kp}_{ss}=\left(\frac{C_{tissue}}{C_{plasma}}\right)}_{ss}={Kp}_{tissue}\cdot \left(1\;-\;{ER}_{tissue}\right)$$
(3)$$V_{dss}=V_{plasma}+\sum_{tissue}\left(V_{tissue}\cdot {Kp}_{tissue}\cdot \left(1\;-\;{ER}_{tissue}\right)\right)$$
where ER is the drug extraction ratio in the metabolic tissues, taking into account the tissue blood flow (Q) and intrinsic clearance (CLint) as ER = CLint/(Q + CLint)). The detailed derivation was provided by Berezhkovskiy in 2010 [31] and more recently revisited by Jeong and Jusko [32]. Additionally, a similar equation was proposed by Bernareggi and Rowland about three decades ago [33]. Moreover, the GastroPlus PBPK platform has implemented this equation to predict the Vdss [13]. These traditional derivations indicate that, after incorporating tissue metabolism into the Vdss, increasing tissue metabolism will decrease the Vdss.

However, the derivations above do come with limitations. First, the above equation for the Kpss is suitable for an isolated perfusion-limited tissue with metabolism. But, how will the Kpss and Vdss be influenced when metabolism occurs in the circulating blood or in the tissue residual blood? Second, it ignores the impact of tissue metabolism on the plasma concentration (Cplasma). Therefore, it deviates from the anatomic physiological reality where all metabolic and non-metabolic tissues are interconnected via circulating plasma. In fact, in a simulation using the Simcyp perfusion-limited PBPK model with hepatic metabolism alone, the tissue/plasma concentration ratio (Kpss) at a pseudo-steady state in non-metabolic tissue is higher than the constant Kptissue used in the above Equation (1). Further, the Vdss derived from the pseudo-steady-state Kpss is higher than the Vdss defined by the constant Kptissue (an example from Simcyp simulation is given in the Appendix A). These observations were results from the hepatic metabolism and plasma-mediated redistribution of drugs from non-metabolic tissues to metabolic tissue. In other words, even with the perfusion-limited PBPK model, tissue metabolism does not always decrease the pseudo-steady-state Kp and Vdss, in contrast to what traditional derivations suggest [13,31,33].

The purpose of the current work was to use a permeability-limited PBPK model to theoretically explore the complex interplay of passive permeability (expressed as the product of the permeability coefficient and surface area, PS), tissue and blood metabolism (CLmet), active uptake or efflux transporters, and dose administration routes (intravenously (IV)/orally (PO)) on the dynamics of the Kp and Vd at a steady state following a continuous IV infusion or at a pseudo-steady state following an instantaneous IV injection or PO absorption. The permeability-limited PBPK model was developed and used to simulate various hypothetical scenarios. To assess the dynamics of the Kp and Vd, the concentration ratio between tissue and venous plasma, the ratio of the amount of drug in the body (including plasma, blood cells, and tissues) to the concentration in the venous plasma were calculated. The following questions will be answered:(1)How will the Kp and Vd differ in the permeability-limited PBPK model compared to those in the perfusion-limited PBPK model?(2)How will transporters impact the Kp and Vd in the permeability-limited PBPK model?(3)Does tissue metabolism decrease the Kp and Vd in the permeability-limited PBPK model?(4)How will the dosing route (IV with bolus or infusion, and PO with fast or slow absorption rate) impact the Kp and Vd?

## 2. Results

### 2.1. PBPK Model

The permeability-limited PBPK model was developed in MATLAB/SimBiology (2022a, Natick, MA, USA). The structure of the permeability-limited PBPK model is shown in Figure 1. Each of the tissue compartments in the permeability-limited PBPK model consists of four subcompartments, namely the tissue blood cells (TCs), tissue plasma (TP), tissue extracellular water (EW), and tissue intracellular water (IW), as shown in Figure 2. Details of the PBPK model (including the model structure, assumptions, governing equations, and system- and drug-specific parameters) and what-if scenario simulations are described in the Materials and Methods. The base model is provided as a Appendix A.

For the sake of simplification, a density of 1 Kg/L is assumed for all tissues and blood. It is further assumed in the what-if simulations that there is no binding or ionization occurring in tissues and blood. With this simplification, Kp = 1 for all tissues and Vdss = 1 L/kg, based on the conventional in silico Kp and Vdss prediction methods. Therefore, if the calculated Kp(t) ≠ 1 or Vd(t) ≠ 1 L/kg in the permeability-limited PBPK model, their differences from one indicate the differences between the permeability-limited PBPK model and the perfusion-limited PBPK model.

### 2.2. Dynamic Kp and Vd in a Closed System

The PK profiles of the tissue and plasma concentrations after an IV bolus administration to an ideal closed system without any metabolism or transporters are shown in Figure 3a. After IV bolus, some tissues (lung and kidney) respond quickly, while some tissues (adipose, muscle, bone, and skin) respond slowly. The concentration in the venous plasma is higher than those in most tissues at the beginning due to the IV bolus administration, but it may be lower than some tissues later due to redistribution. However, because it is a closed system, the same constant concentration will eventually be achieved for all tissues and plasma.

The simulated Kp profiles are shown in Figure 3b. After IV bolus to a closed system without metabolism or transporters, the fast-responding tissues (lung and kidney) reach the steady state with Kp = 1 quickly, while the slow-responding tissues (adipose, muscle, bone, and skin) have Kp ≠ 1 at the beginning, though their final steady-state Kp = 1 as well. It takes time for those slow-responding tissues to reach equilibrium (steady state). The dynamics of the tissue Kp is defined by the tissue volume and its blood flow.

The dynamics of the Vd are given in Figure 3c. After IV bolus, without metabolism or transporters in the closed system, passive permeability (PS) has significant impact on the Vd in reaching Vdss = 1 L/kg. High permeability (PS > 1-fold that of Qtissue) enables Vd to reach 1 L/kg quickly, but 5 h seems to be the minimal time needed, even with high passive permeability (Figure 3c), where the Vd profiles are overlapped for PS/Q = 10 and PS/Q = 100.

### 2.3. Dynamic Kp and Vd in an Open System

The PK profiles of the tissue and plasma concentrations after an IV bolus administration to an open system with hepatic metabolism are shown in Figure 4a. After IV bolus, with linear metabolism in the liver, the liver has the lowest concentration. The concentration in the venous plasma is lower than those in all of the non-metabolic tissues, while the concentrations in adipose, muscle, bone, and skin are relatively higher than the other tissues. All tissue and plasma concentrations will be at a pseudo-steady state when the redistribution and elimination are in equilibrium and all of the PK profiles have the same slope (i.e., the same elimination half-life) in the semi-log plot.

In contrast to the closed system where Kp = 1 after reaching a steady state, the open system has Kp < 1 in the metabolic tissue (liver) but Kp > 1 in all of the non-metabolic tissues, as shown in Figure 4b. After IV bolus with metabolism in the liver, the fast-responding tissues (lung Kpss = 1.0005 and kidney Kpss = 1.003) reach their pseudo-steady-state Kpss quickly, and the slow-responding tissues (adipose, muscle, bone, and skin) reach their pseudo-steady state slowly. Adipose has the highest Kp, which may significantly impact the dynamics of the Vd.

The dynamics of the Vd are shown in Figure 4c. Compared to the closed system where the Vdss = 1 L/kg after reaching a steady state, after IV bolus to an open system with hepatic metabolism, the Vdss > 1 L/kg. Hepatic metabolism has significant impact on the pseudo-steady-state Vdss; namely, increasing hepatic metabolism will increase the Vdss. This is fundamentally in contrast to the derivations from the traditional perfusion-limited PBPK model [13,31,33].

### 2.4. Impact of Passive Permeability

The impact of passive permeability on the Vdss from different what-if scenario simulations in the open system with metabolic clearance in the intracellular water of all tissues is summarized in Figure 5 and Table 1.

For IV infusion and PO slow absorption, increasing permeability will consistently increase the Vdss, though the magnitude is very small. Increased permeability enables more of a drug to distribute into the intracellular water, resulting in a higher Vdss.

For IV bolus administration and PO fast absorption, increasing permeability will markedly increase the Vdss when the PS is less than the tissue blood flow (Q), though the magnitude of the increase will be saturated when the PS is more than 1-fold that of Q.

If the PS is much smaller than the Q (0.01-fold of PS/Q, shown in Table 1), the Vd is independent of the various scenarios tested, consistent with less tissue distribution for drugs of low passive permeability.

### 2.5. Impact of Metabolism

#### 2.5.1. Hepatic Metabolism Only

The impact of metabolism on Vdss from different what-if scenario simulations in the open system with hepatic metabolic clearance only is summarized in Figure 6 and Table 2. Hepatic metabolism from any liver subcompartment has a big impact on Vdss for IV bolus administration and PO fast absorption. Compared to metabolism in the tissue (IW and EW), metabolism in the residual blood (plasma or blood cells) has a bigger impact on Vdss.

Only for IV infusion, increasing tissue metabolism in any subcompartment will decrease the tissue Kp and result in a decreased Vdss. This is consistent with the traditional derivations from the perfusion-limited PBPK model [13,31,33].

For IV bolus administration or any PO absorption, increasing tissue metabolism will increase the tissue Kp and therefore increase the Vdss, though the increase is minor in the case of PO slow absorption. This is fundamentally in contrast to the conclusions from Berezhkovskiy [31] and Bernareggi and Rowland [33].

#### 2.5.2. Metabolism in All Tissues

The impact of metabolism on the Vdss from different what-if scenario simulations in the open system with metabolic clearance in different tissue subcompartments is summarized below in Figure 7 and Table 3. Like the previous case of hepatic metabolism alone, metabolism has a large impact on the Vdss for IV bolus administration and PO fast absorption, but not for IV infusion and PO slow absorption.

For IV bolus administration and PO fast absorption, increasing tissue metabolism will result in an increased Vdss, in contrast to the conclusions from Berezhkovskiy [31] and Bernareggi and Rowland [33].

Further for IV bolus and PO fast absorption, metabolism in the residual blood (plasma or blood cells) has a larger impact on the Vdss compared to metabolism in the tissue (IW or EW).

### 2.6. Impact of Transporters

#### 2.6.1. Closed System

The impact of transporters on the Vd in a closed system after an IV bolus administration is shown in Figure 8. After IV bolus without metabolism in the closed system, transporters have a significant impact on the Vd of low-permeability compounds. Uptake transporters increase the Vd and delay the Vd to reach Vdss, while efflux transporters decrease the Vd. Compared to efflux transporters, uptake transporters have a larger impact on the Vd, regardless of the permeability, though the difference would be more obvious for low-permeability drugs.

#### 2.6.2. Open System

The impact of transporters on the Vdss from different what-if scenario simulations in the open system with metabolic clearance in all tissues is summarized in Figure 9 and Table 4. As is well-recognized, an uptake transporter will enhance the accumulation of its substrate from circulatory blood into the tissue, while an efflux transporter will enhance the decumulation of its substrate from the tissue to circulation. As a result, compared to no transporter involvement, uptake transporters increased the Kp and Vdss, while efflux transporters decreased the Kp and Vdss.

It was observed that for compounds with low permeability, active transporters have a large impact on the Vdss.

### 2.7. Impact of Dosing in an Open System

For IV infusion, a real steady state can be achieved if the infusion rate and elimination rate are balanced. Regardless of if metabolism or elimination exists in the liver alone or in all tissues, or if the compound has high or low permeability, increasing metabolism will decrease the Kp in metabolic tissues while keeping the Kp = 1 in non-metabolic tissues. As a result, a decrease in Vdss is expected, in agreement with the derivations by Berezhkovskiy using the traditional perfusion-limited PBPK model [13,31,33].

It was observed, remarkably, that the results from IV bolus administration and those from PO fast absorption were identical with regard to the impact of passive permeability, metabolism in tissue, and active transporters on the Vdss, although the dynamics of the Kp and Vd were different. After IV bolus administration, the concentration in the venous plasma will always be higher than the concentration in any tissue in the moments immediately after dosing, regardless of metabolism. For PO fast absorption, the gut concentration could be high at the beginning immediately after PO administration. However, for both IV bolus administration and PO fast absorption, the same quasi-steady state will be achieved at the elimination phase. Therefore, the calculated Vdss at a quasi-steady state will be the same.

However, the impact of PO slow absorption on the Vdss is not conclusive, mostly due to the flip-flop PK in this case.

### 2.8. Global Sensitivity Analysis

A global sensitivity analysis indicated that tissue metabolism, passive permeability, and active transporters had different contributions to the Vdss, depending on the dosing routes (Figure 10). For IV infusion where a steady state was achievable, passive permeability between plasma and extracellular water had the most impact on the Vdss, followed by intracellular metabolism. For IV bolus, passive permeability and efflux transporters had higher contributions to the Vdss, compared to subcompartment metabolism and uptake transporters. PO fast absorption demonstrated a similar trend, although the contribution of passive permeability between plasma and extracellular water was minor. For PO slow absorption, passive permeability and efflux transporters had major contributions to the Vdss.

In general, tissue metabolism contributed less to the Vdss than passive permeability and active transporters when these parameters change 10-fold from their baselines that were defined as 10% of the tissue blood flow.

## 3. Discussion

The Kp and Vdss are two essential parameters in the traditional perfusion-limited PBPK model. Over the past 50 years, various mechanistic methods have been developed to predict them, but enhancing their predictive accuracy has remained a significant challenge for both academic researchers and industrial leaders [17,18,19,20,21,22,23,24,25]. While reviewing the mechanistic prediction of the Kp and Vdss is beyond the scope of the current work, below are some cornerstones in the methodology’s development based on our knowledge:(1)Poulin and Krishnan developed the first biologically based algorithm to predict tissue/blood partition coefficients based on the ratio of the solubility (rather than binding) in tissue and that in blood [34,35]. The solubility in tissue or in blood was estimated by considering the total solubility of a drug in neutral lipids, phospholipids, and water present in tissue or in blood.(2)Poulin and colleagues further extended their work to predict the Vdss by treating the human body as being composed of adipose and non-adipose tissues. They used the n-octanol/water partition coefficient and the vegetable oil/water distribution coefficient to approximate the hydrophobic interactions of drugs with the biologic lipids of adipose and non-adipose tissues, respectively [22,23,36,37]. The ‘Poulin–Theil method’ implicitly assumes that the tissue is homogeneous without cell membranes, and all water, neutral lipids, and phospholipids are evenly distributed in extracellular and intracellular spaces.(3)Berezhkovskiy derived Poulin–Theil’s equation based on the same assumption but made a correction to the Poulin–Theil method by considering the difference in water and phospholipid fractions in tissue vs. plasma [38].(4)Rodgers and Rowland developed a set of mechanistic equations for Kp and Vdss prediction based on more detailed tissue composition and compound types [24,25,39]. In detail, different tissues have different fractions of water, pH, lipids, and proteins in the intracellular and extracellular spaces, separated by tissue cell membranes. It is assumed in the ‘Rodgers–Rowland method’ that all drugs dissolved in intracellular and extracellular tissue water follow the classical pH-partition hypothesis across cell membranes and differentiate into the neutral lipids and neutral phospholipids located within the intracellular water. For compounds with basic pKa ≥ 7 (ionized bases and positively charged zwitterions), electrostatic interaction with intracellular acidic phospholipids is incorporated, while acids/weakly basic or neutral compounds are assumed to bind primarily to extracellular albumin or neutral lipoproteins, respectively.

The Poulin–Theil, Berezhkovskiy, and Rodgers–Rowland methods have all been implemented in GastroPlus [13] and Simcyp [14] and adopted by many bespoke PBPK models [40,41,42]. When using these in silico methods to predict Kp and Vdss values and further applying them to perfusion-limited PBPK modeling and simulation, the first and the most important assumption is that the system and PK are at a steady state, which can only be achievable in the following:(1)A closed system with a constant tissue concentration and constant plasma concentration after an instantaneous dose administration, or(2)An open system with a constant drug uptake (such as IV infusion) balanced through constant elimination; therefore, the tissue concentration and plasma concentration are in equilibrium.

However, the human body is an open system, and constant IV infusion is rare. After an instantaneous dose administration to an open system with net metabolism, the concentration in plasma and that in the tissues are neither constant nor at a steady state. At most, there might be pseudo equivalence between the blood and the tissues at a pseudo-steady state.

On the other hand, tissue metabolism is believed to decrease the Kp and Vdss via the extraction ratio (ER) [13,33]. Berezhkovskiy gave a detailed derivation using the traditional perfusion-limited PBPK model [31]. However, Berezhkovskiy’s derivation was based on the ordinary differential equation (ODE) for the metabolic tissue alone, and the impact of metabolism on the concentrations of plasma and non-metabolic tissues was not incorporated. When there is tissue metabolism in an open system, the concentration will be high in the non-metabolic tissues but low in the metabolic tissues, and the plasma concentration is between them. Therefore, tissue metabolism is expected to decrease the Kp for the metabolic tissues but to increase the Kp for the non-metabolic tissues; hence, the Vdss may not simply decrease.

Even for a constant Kp used in the traditional perfusion-limited PBPK model, the calculated Ctissue/Cplasma has dynamics. Lin and colleagues introduced an S-shaped time-varying distribution coefficient to replace the constant Kp in the traditional perfusion-limited PBPK model to describe the experimental PK of nanoparticles after IV bolus administration to mice [43]. As observed in the simulations with a permeability-limited PBPK model (Figure A1, PO fast absorption, CL_liverIW = 0.1Q), except for the gut (where the oral dose is given) and the liver (where metabolism exists), most of the other tissues have an S-shaped Kp. Such Kp changes between 0.1 and 20 h post PO administration in most of the tissues highlight the potential risk of using the perfusion-limited PBPK model with a constant Kp in analyzing clinical PK data. Future work remains to predict tissue-specific S-shaped distribution coefficient functions (such as via Emax, EC50, and Hill coefficients) mechanistically.

It is worth noting that the traditional in silico methods for predicting the Kp and Vdss did not take into account the role of active transporters in the drug disposition within tissues in the past. As the PBPK field advances, it becomes increasingly important to consider more complicated and physiologically relevant scenarios, including tissue transporters. It is expected that an uptake transporter would increase the tissue Kp, as implemented in Simcyp for hepatic clearance [14]. The current permeability-limited PBPK model has confirmed this concept. After IV bolus, without metabolism in the closed system, transporters have significant impact on the Vd of low-permeability compounds. Uptake transporters increase the Vd and delay the Vd to reach the Vdss, while efflux transporters decrease the Vd. Compared to efflux transporters, uptake transporters have a larger relative impact on the Vd, regardless of the permeability, though the difference would be more obvious for a low-permeability drug. The same holds for an open system with IV infusion or PO absorption.

The current work systematically explored the interplay of passive permeability, metabolism in tissue or residual blood, efflux or uptake transporters, and various dosing routes in determining the dynamics of the Kp and Vd with a novel permeability-limited PBPK model. Clearly, permeability is a key parameter in defining drug disposition within tissues. For a low-permeability drug, the tissue intracellular concentration may significantly differ from the extracellular concentration and the plasma concentration; therefore, it is hard to justify the assumption of a homogenous tissue with similar free drug concentrations across the tissue blood plasma, extracellular, and intracellular spaces.

The conventional in silico methods for Kp and Vdss prediction have assumed steady states, and their robustness under different dynamic dosing scenarios has not been assessed systematically. The impact of different dosing strategies on the Kp and Vdss were formally explored here. The Kp profiles generated by the permeability-limited PBPK model for IV bolus administration and PO fast absorption are always greater than those generated for IV infusion or PO slow absorption. It was observed, remarkably, that the results from IV bolus administration and those from PO fast absorption were the same, with regard to the impact of passive permeability, metabolism in tissue, and active transporters on the Vdss, although the dynamics of the Kp and Vd were different. Immediately after IV bolus administration, the concentration in the venous plasma will always be higher than the concentration in any tissue, regardless of metabolism. For PO fast absorption, the gut concentration could be high at the beginning, immediately after PO administration. However, for both IV bolus administration and PO fast absorption, the same quasi-steady state will be achieved at the elimination phase. Therefore, the calculated Vdss at a quasi-steady state will be the same.

Regardless of if metabolism exists in the liver alone or in all tissues, or whether the compound has high permeability or low, tissue metabolism results in a decreased Kp in metabolic tissues but not in non-metabolic tissues. In IV bolus administration and PO fast absorption in the open system, tissue metabolism will increase the Vdss, in contrast to the traditional derivation. It is not conclusive for PO slow absorption, mostly due to the flip-flop kinetics involved. Another striking observation was the time scale and magnitude of deviation from the perfusion-limited model in several of the presented test scenarios. For example, in the IV bolus simulation scenario with tissue metabolism, the dynamic adipose tissue Kp value did not stabilize until 30 h following administration, and the Kp value was markedly greater than 1 (Figure 4).

Below are some limitations of the current work:(1)The current model considered linear transport and metabolism; namely, non-saturable kinetics is assumed for transporters and metabolism in tissues and blood. Further mechanistic fidelity can be added by incorporating concentration-dependent transport and metabolism considerations, such as saturation.(2)The passive permeability across cell membranes is based on the classic pH-partition hypothesis; namely, only free (unbound and unionized) drugs can passively permeate through the biological membrane. However, it has been proposed that both ionized drugs and bound drugs can penetrate the cell membrane depending on the membrane potential or specific carrier mechanisms [44,45,46].(3)Lymphatic circulation is not considered in the current model. However, for large molecules and PO slow absorption, lymph may have a significant impact on drug disposition.(4)The current four-subcompartment tissue may be further expanded to incorporate endothelium and organelles (such as lysosomes and mitochondrion), such that their contribution to tissue disposition can be considered in the PBPK model.(5)Protein binding and ionization have been ignored in all what-if simulations for the sake of simplification and to enable quick interpretation. Tissue composition and pH may significantly affect the local binding and ionization in each subcompartment.(6)Slow oral absorption with Ka = 0.01 1/h may be too simplistic to describe the flip-flop PK due to extended/controlled formulations. Further mechanistic oral absorption models may help to highlight the impact of slow absorption on tissue disposition.

## 4. Materials and Methods

### 4.1. Model Structure

Like all traditional perfusion-limited PBPK models, the permeability-limited PBPK model approximates the human body by using a series of tissue compartments connected through the blood circulation, as shown in Figure 1.

However, different from the traditional PBPK of well-stirred perfusion-limited tissue compartments, each of the tissue compartments in Figure 1 consists of four subcompartments, namely tissue blood cells (TCs), tissue plasma (TP), tissue extracellular water (EW), and tissue intracellular water (IW), as shown in Figure 2. Such a permeability-limited four-subcompartment structure has been applied previously to investigate the disposition of cyclosporine in rats and humans [47,48].

The tissue blood cells (TCs) and plasma (TP) correspond to the residual blood within the tissue, which can be defined based on rat data [33]. The extracellular water (EW) subcompartment and the intracellular water (IW) subcompartment are defined as implemented in Simcyp Simulator (V.21, Sheffield, UK) [14]. For all blood compartments, including arterial blood, venous blood, and portal vein blood, there is no extracellular or intracellular water, and only the blood cell and plasma subcompartments exist. In addition, blood flow consists of plasma flow (Qtp) and blood cell flow (Qtc).

There is passive permeation between the adjacent subcompartments, as well as an active uptake and efflux across the cell membrane between the extracellular and intracellular water. In general, each of the subcompartments is assumed to be well stirred, while drug transport between adjacent subcompartments is intrinsically permeability-limited, though it can be reduced into a well-stirred compartment by assuming a high value of all the products of the passive permeability coefficient surface area (PS) between the adjacent compartments.

Note that active efflux or uptake transporters have been considered on the tissue cell membrane only while metabolism can occur in any subcompartment.

### 4.2. Governing Equations

#### 4.2.1. Tissue Subcompartment Concentration

Based on mass balance, various ordinary differential equations (ODEs) can be used to describe the concentration kinetics in each of the subcompartments in tissue and blood compartments in the permeability-limited PBPK model.

##### Typical Tissue Receiving the Arterial Blood Perfusion

These tissues include adipose, bone, brain, heart, kidney, muscle, skin, pancreas, spleen, and gut tissue (Figure 1).

Intracellular water (iw) subcompartment:


(4)
$$V_{iw}\frac{dC_{iw}}{dt}=J_{ew↔iw}{+J}_{ew\to iw}{\;-\;J}_{iw\to ew}\;-\;J_{iw\to met}$$


Extracellular water (ew) subcompartment:


(5)
$$V_{ew}\frac{dC_{ew}}{dt}=J_{tp↔ew}\;-\;J_{ew↔iw}{\;-\;J}_{ew\to iw}{+J}_{iw\to ew}\;-\;J_{ew\to met}$$


Tissue plasma (tp) subcompartment:


(6)
$$V_{tp}\frac{dC_{tp}}{dt}=Q_{tp}\left(C_{ap}\;-\;C_{tp}\right)+J_{tc↔tp}\;-\;J_{tp↔ew}\;-\;J_{tp\to met}$$


Tissue BC (tc) subcompartment:

(7)$$V_{tc}\frac{dC_{tc}}{dt}=Q_{tc}\left(C_{ac}\;-\;C_{tc}\right)\;-\;J_{tc↔tp}\;-\;J_{tc\to met}$$
where V, C, t, and J are the volume (in unit of L), concentration (mg/L), time (h), and mass flux (mg/h), respectively. Qtp and Qtc are the plasma flow rate (L/h) and blood cell flow rate (L/h) to the target tissue. The arterial plasma and arterial blood cells are indicated as ap and ac. Note that the bidirectional arrow (↔) in the subscript describes the passive permeation between the adjacent subcompartments, whereas the unidirectional arrow (→) describes the active transport from one subcompartment to its adjacent subcompartment.

Based on the pH-partition hypothesis, only unbound and unionized drugs can passively penetrate across biological membranes. Therefore, various bidirectional passive mass fluxes (J) are defined below:(8)$$J_{ew↔iw}={PS}_{ew/iw}\left(C_{ew}\cdot {fu}_{ew}\cdot {fi}_{ew}\;-\;C_{iw}\cdot {fu}_{iw}\cdot {fi}_{iw}\right)$$
(9)$$J_{tp↔ew}={PS}_{tp/ew}\left(C_{tp}\cdot {fu}_{tp}\cdot {fi}_{tp}\;-\;C_{ew}\cdot {fu}_{ew}\cdot {fi}_{ew}\right)$$
(10)$$J_{tc↔tp}={PS}_{tc/tp}\left(C_{tc}\cdot {fu}_{tc}\cdot {fi}_{tc}\;-\;C_{tp}\cdot {fu}_{tp}\cdot {fi}_{tp}\right)$$

The unidirectional mass flux (J) via activate transporters are defined below:(11)$$J_{ew\to iw}={CL}_{ew/iw}\cdot C_{ew}\cdot {fu}_{ew}$$
(12)$$J_{iw\to ew}={CL}_{iw/ew}\cdot C_{iw}\cdot {fu}_{iw}$$

The metabolic mass flux (J) are defined below:(13)$$J_{iw\to met}={CL}_{iw/met}\cdot C_{iw}\cdot {fu}_{iw}$$
(14)$$J_{ew\to met}={CL}_{ew/met}\cdot C_{ew}\cdot {fu}_{ew}$$
(15)$$J_{tp\to met}={CL}_{tp/met}\cdot C_{tp}\cdot {fu}_{tp}$$
(16)$$J_{tc\to met}={CL}_{tc/met}\cdot C_{tc}\cdot {fu}_{tc}$$
where PS (L/h) is the product of the passive permeability coefficient (P) and the surface area (S) between the adjacent subcompartments; fu and fi are the unbound fraction and the unionized fraction in each subcompartment. CLew/iw and CLiw/ew are the clearance (L/h) mediated by active transporters from the extracellular water to the intracellular water, and vice versa, from the intracellular water to the extracellular water. CLiw/met, CLew/met, CLtp/met, and CLtc/met are the metabolic clearance (L/h) in the each subcompartment.

##### Lung

For lung tissue, which is perfused by venous plasma and venous blood cells, the following 4 ODEs are used:Intracellular water (iw) subcompartment:
(17)$$V_{iw}\frac{dC_{iw}}{dt}=J_{ew↔iw}{+J}_{ew\to iw}{\;-\;J}_{iw\to ew}\;-\;J_{iw\to met}$$

Extracellular water (ew) subcompartment:


(18)
$$V_{ew}\frac{dC_{ew}}{dt}=J_{tp↔ew}\;-\;J_{ew↔iw}{\;-\;J}_{ew\to iw}{+J}_{iw\to ew}\;-\;J_{ew\to met}$$


Tissue plasma (tp) subcompartment:


(19)
$$V_{tp}\frac{dC_{tp}}{dt}=Q_{vp}\left(C_{vp}\;-\;C_{tp}\right)+J_{tc↔tp}\;-\;J_{tp↔ew}\;-\;J_{tp\to met}$$


Tissue BC (tc) subcompartment:

(20)$$V_{tc}\frac{dC_{tc}}{dt}=Q_{vc}\left(C_{vc}\;-\;C_{tc}\right)\;-\;J_{tc↔tp}\;-\;J_{tc\to met}$$
where Qvp and Qvc are the total venous plasma flow rate (L/h) and the total venous blood cell flow rate (L/h) from the venous blood to the lung. Their sum is the cardiac output.

##### Liver

For the liver, which is perfused by hepatic arterial (ha) blood and portal venous (pv) blood, the following 4 ODEs are used:Intracellular water (iw) subcompartment:
(21)$$V_{iw}\frac{dC_{iw}}{dt}=J_{ew↔iw}{+J}_{ew\to iw}{\;-\;J}_{iw\to ew}\;-\;J_{iw\to met}$$

Extracellular water (ew) subcompartment:


(22)
$$V_{ew}\frac{dC_{ew}}{dt}=J_{tp↔ew}\;-\;J_{ew↔iw}{\;-\;J}_{ew\to iw}{+J}_{iw\to ew}\;-\;J_{ew\to met}$$


Tissue plasma (tp) subcompartment:


(23)
$$V_{tp}\frac{dC_{tp}}{dt}=Q_{ha\_p}C_{hap}+Q_{pv\_p}C_{pvp}\;-\;Q_{liver\_p}C_{tp}+J_{tc↔tp}\;-\;J_{tp↔ew}\;-\;J_{tp\to met}$$


Tissue BC (tc) subcompartment:

(24)$$V_{tc}\frac{dC_{tc}}{dt}=Q_{ha\_c}C_{hac}+Q_{pv\_c}C_{pvc}\;-\;Q_{liver\_c}C_{tc}\;-\;J_{tc↔tp}\;-\;J_{tc\to met}$$
where Qha_p and Qha_c are the hepatic arterial plasma flow (L/h) and blood cell flow (L/h), while Qpv_p and Qpv_c are the portal venous arterial plasma flow (L/h) and blood cell flow (L/h), respectively. The following flow rate balances are maintained:(25)$$Q_{liver\_p}=Q_{ha\_p}+Q_{pv\_p}$$
(26)$$Q_{liver\_c}=Q_{ha\_c}+Q_{pv\_c}$$

##### Portal Vein

The portal vein is a blood compartment, consisting of plasma and blood cell subcompartments:Portal venous plasma (pvp) subcompartment:
(27)$$V_{pvp}\frac{dC_{pvp}}{dt}=Q_{pancreas\_p}C_{panceas\_p}+Q_{spleen\_p}C_{spleen\_p}+Q_{gut\_p}C_{gut\_p}\;-\;Q_{pvp}C_{pvp}+J_{pvc↔pvp}\;-\;J_{pvp\to met}$$

Portal venous blood cell (pvc) subcompartment:

(28)$$V_{pvc}\frac{dC_{pvc}}{dt}=Q_{pancreas\_c}C_{panceas\_c}+Q_{spleen\_c}C_{spleen\_c}+Q_{gut\_c}C_{gut\_c}\;-\;Q_{pvc}C_{pvc}\;-\;J_{pvc↔pvp}\;-\;J_{pvc\to met}$$
where Qpancreas_p, Qspleen_p, and Qgut_p are the plasma flow (L/h) to the portal venous plasma subcompartment from the pancreas, spleen, and gut, respectively, while Qpancreas_c, Qspleen_c, and Qgut_c are the blood cell flow (L/h) to the portal venous blood cell subcompartment from the pancreas, spleen, and gut, respectively. The following flow rate balances are maintained:(29)$$Q_{pvp}=Q_{pancreas\_p}+Q_{spleen\_p}+Q_{gut\_p}$$
(30)$$Q_{pvc}=Q_{pancreas\_c}+Q_{spleen\_c}+Q_{gut\_c}$$

##### Venous Blood

Venous plasma (vp) subcompartment:


(31)
$$V_{vp}\frac{dC_{vp}}{dt}=\sum_{tissue}\left(Q_{tp}C_{tp}\right)\;-\;Q_{vp}C_{vp}+J_{vc↔vp}\;-\;J_{vp\to met}$$


Venous blood cell (vc) subcompartment:

(32)$$V_{vc}\frac{dC_{vc}}{dt}=\sum_{tissue}\left(Q_{tc}C_{tc}\right)\;-\;Q_{vc}C_{vc}\;-\;J_{vc↔vp}\;-\;J_{vc\to met}$$
where tissue includes adipose, bone, brain, heart, kidney, muscle, skin, and liver. Qvp and Qvc are the total venous plasma flow rate (L/h) and the total venous blood cell flow rate (L/h) from the venous blood to the lungs. Their sum is the cardiac output.
(33)$$Q_{vp}=Q_{adipose\_p}+Q_{bone\_p}+Q_{brain\_p}+Q_{heart\_p}+Q_{kidney\_p}+Q_{muscle\_p}+Q_{skin\_p}+Q_{liver\_p}$$
(34)$$Q_{vc}=Q_{adipose\_c}+Q_{bone\_c}+Q_{brain\_c}+Q_{heart\_c}+Q_{kidney\_c}+Q_{muscle\_c}+Q_{skin\_c}+Q_{liver\_c}$$

##### Arterial Blood

Arterial plasma (ap) subcompartment:


(35)
$$V_{ap}\frac{dC_{ap}}{dt}=Q_{vp}\left(C_{lung\_p}\;-\;C_{ap}\right)+J_{ac↔ap}\;-\;J_{ap\to met}$$


Arterial blood cell (ac) subcompartment:


(36)
$$V_{ac}\frac{dC_{ac}}{dt}=Q_{vc}\left(C_{lung\_c}\;-\;C_{ac}\right)\;-\;J_{ac↔ap}\;-\;J_{ac\to met}$$


#### 4.2.2. Dose Administration

For IV bolus or IV infusion, a dosing term is applied to the above ODE (Equation (31)) for the venous plasma compartment. For PO, the following dosing term is applied to the above ODE (Equation (4)) for the intracellular water subcompartment of the gut tissue:(37)$$Dosing=Dose\cdot K_{a}{\cdot e}^{\;-\;K_{a}\cdot t}$$
where Dose is the amount (mg) administrated and Ka is the absorption rate constant (1/h). Ka is set high or low to describe fast or slow oral absorption in the what-if scenario simulations.

#### 4.2.3. Calculation of Kp and Vdss

##### Tissue Concentration

The tissue concentration (mg/L) is defined by the total amount (mg) in the tissue and the total volume (L) of the tissue, as given below:(38)$$C_{tissue}(t)=\frac{V_{iw}C_{iw}+V_{ew}C_{ew}+V_{tp}C_{tp}+V_{tc}C_{tc}}{V_{iw}+V_{ew}+V_{tp}+V_{tc}}$$

##### Tissue/Plasma Partition Coefficient

The tissue/plasma partition coefficient is defined below, using the venous plasma concentration as the surrogate of the systemic plasma concentration:(39)$$Kp(t)=\frac{C_{tissue}(t)}{C_{vp}}$$

##### Volume of Distribution

The volume of distribution (Vd, L/kg) is defined by the ratio of the amount (mg) in the body and the venous plasma concentration (mg/L), further normalized by body weight (BW, kg):(40)$$Vd\left(t\right)=\frac{V_{vp}C_{vp}+V_{vc}C_{vc}+V_{ap}C_{ap}+V_{ac}C_{ac}+\sum_{tissue}\left(V_{tissue}C_{tissue}\right)}{C_{vp}\cdot BW}$$

The Vdss was taken from the simulation when the pseudo-steady state was achieved.

### 4.3. Model Parameterization

#### 4.3.1. System-Specific Data

Different organs have different densities [49]. For the sake of simplification, a density of 1 kg/L is assumed for all tissues and blood. Therefore, the sum of tissue and blood volumes will be numerically equal to the BW. If the drug is evenly distributed in all plasma, blood cell, and tissue subcompartments, then Kp = 1 for all tissues and Vdss = 1 L/kg.

In the current PBPK modeling exercise, the distribution of the tissue volume and blood flow are based on Simcyp Simulator [14], ICRP [50], and Brown et al. [49], as shown in Table A1, with minor adjustments to make a 100% balance of the total body weight (70 kg) and cardiac output (300 L/h) with the tissues shown in Figure 1.

As mentioned previously, the residual blood volume is based on the data of blood remaining in rat tissues after bleeding [33], and the volume of residual blood is further separated into tissue plasma (TP) and blood cell (TC) subcompartments, based on the hematocrit; namely, 45% of residual blood is blood cells and the remaining 55% is the residual plasma.

#### 4.3.2. Compound-Specific Data

The following drug-specific data are used for this exercise:PS = 0 or comparable to Qtissue in each tissue;Clmet = 0 or comparable to Qtissue in each tissue;Cluptake = 0 or comparable to Qtissue in each tissue;Clefflux = 0 or comparable to Qtissue in each tissue.

If the metabolism Clmet = 0, the system is a closed system. Otherwise, it is an open system.

For the sake of simplification, it is assumed that there is no binding or ionization; therefore, fu_tc = fu_tp = fu_ew = fu_iw = 1, and fi_tc = fi_tp = fi_ew = fi_iw = 1 in all tissues and blood. With this simplification, Kp = 1 for all tissues and Vdss = 1 L/kg, based on the conventional in silico Kp and Vdss prediction methods. Therefore, if the calculated Kp(t) ≠ 1 or Vd(t) ≠ 1 L/kg in the permeability-limited PBPK model, their differences from 1 indicate the differences between the permeability-limited PBPK model and the perfusion-limited PBPK model.

#### 4.3.3. Dosing

In this PBPK modeling exercise, for the sake of illustration, the following dose regimens were simulated, though no difference from the dose amount was expected in a linear system for the simulated Kp and Vd that are based on the concentration ratio of tissues and plasma:(1)Constant IV infusion at 100 mg/h to the open system;(2)Bolus IV injection of 100 mg to the closed system or the open system;(3)Oral absorption of 100 mg with Ka = 1 1/h for fast oral absorption or Ka = 0.01 1/h for slow oral absorption into an open system.

### 4.4. What-If Scenarios

#### 4.4.1. Impact of Passive Permeability

Passive permeability exists between all of the adjacent subcompartments. The impact of passive permeability was explored by varying the PS, the product of passive permeability coefficient (P), and the surface area (S) around the tissue blood flow.

For a typical small molecule, the permeability coefficient on the cell membranes will generally be smaller than that on the vascular membrane. Further, assuming an immediate equilibrium between the tissue residual plasma (TP) and blood cells (TC), the passive permeability between these two subcompartments (PStc/tp) were set to 1000 L/h.

#### 4.4.2. Impact of Metabolism

Metabolism can exist everywhere in the human body. It can be in a specific tissue (such as in the liver due to a specific P450 enzyme) or in all subcompartments including the residual blood (due to hydrolysis).

The impact of metabolism was explored by varying the metabolic clearance (CLmet) around tissue blood flow.

#### 4.4.3. Impact of Transporters

Active uptake and efflux transporters are incorporated on the cell membranes between the extracellular water (EW) and the intracellular water (IW) subcompartments in the current model.

The impact of transporters was explored by varying the uptake or efflux transporter clearance (CLew/iw or CLiw/ew) around the tissue blood flow.

#### 4.4.4. Impact of Dosing Route

The impact of the dosing route was explored by examining the Kp and Vd after IV bolus, IV infusion, and PO with fast (Ka = 1 1/h) or slow (Ka = 0.01 1/h) absorption.

### 4.5. Global Sensitivity Analysis

In addition to the above what-if scenarios, a global sensitivity analysis was applied to the Vd after IV bolus, IV infusion, and PO with fast or slow absorption with the metabolic clearance in each subcompartment, the passive permeability between plasma/EW/IW, and the uptake/efflux transporters. All of these sensitivity inputs were set at 1% of the tissue blood flow as the mean while changing 10-fold higher or lower, and 1000 parameter samples were used to compute the Sobol index.

## 5. Conclusions

Conventional in silico methods to predict the Kp and Vdss were based on theoretical assumptions such as well-stirred tissue without metabolism and no active transporters. A permeability-limited PBPK model with four subcompartments (intracellular and extracellular water, residual plasma, and blood cells) for each tissue has been developed in MATLAB/SimBiology and used to explore the interplay of passive permeability, metabolism, active transporters, and dosing routes in determining the dynamics of the Kp and Vd. Various what-if scenario simulations highlighted that the Kp and Vdss at the pseudo-steady state were fundamentally different from the Kp and Vdss derived from the conventional perfusion-limited PBPK model. Importantly, in an open system with instantaneous drug administration, tissue metabolism may significantly increase the Kp in non-metabolic tissues and result in an increase in the Vdss. The traditional perfusion-limited PBPK model with a constant Kp may not capture many features of tissue drug disposition due to overly simplified assumptions regarding the mechanisms of drug disposition, especially after IV bolus administration or PO fast absorption. The permeability-limited PBPK model has fully integrated multiple biological mechanisms, including passive permeability across cell membranes, metabolism within tissue and residual blood, active uptake and efflux transporters, and complex dosing routes. These advantages provide a strong rationale to widely adopt the permeability-limited PBPK model for describing clinical PK and explaining observed Kp and Vdss values.

## Figures and Tables

**Figure 1 ijms-24-16224-f001:**
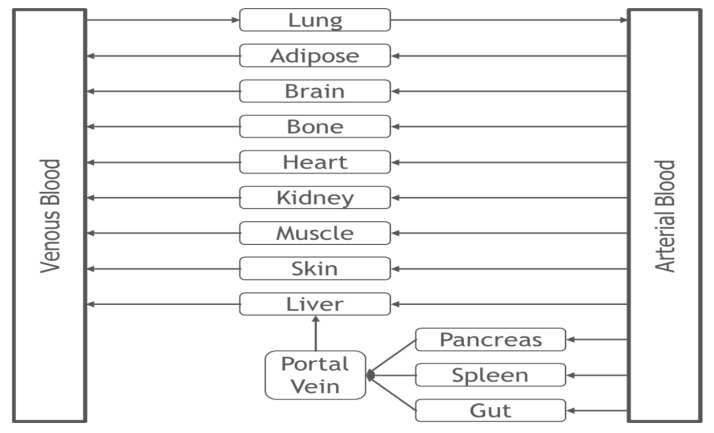
Structure of PBPK model. The PBPK model describes the human body using 12 tissue compartments and 3 blood compartments. All tissue and blood compartments are linked together through the blood circulation. Each tissue compartment is further separated into 4 subcompartments, consisting of tissue blood cells, tissue plasma, and extracellular and intracellular water, as shown in Figure 2. The blood compartments (arterial blood, portal vein, and venous blood) consist of 2 subcompartments, namely blood cells and plasma.

**Figure 2 ijms-24-16224-f002:**
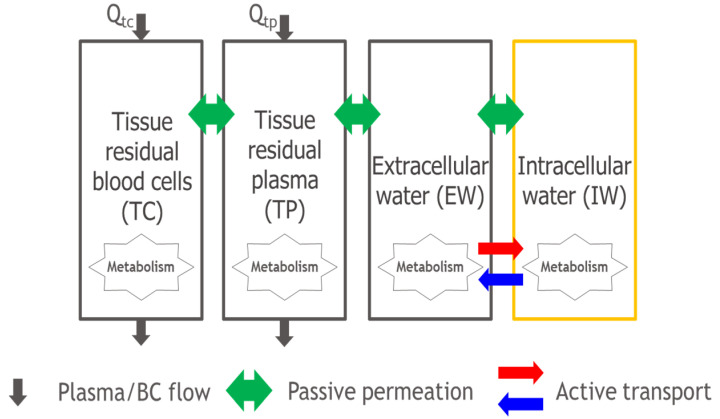
Subcompartments of permeability-limited tissue, TCs, TP, EW, and IW are tissue residual blood cells, tissue residual plasma, extracellular water, and intracellular water, respectively. Qtp and Qtc are plasma flow rate (L/h) and blood cell flow rate (L/h) to the target tissue. Bidirectional arrow is the passive permeation between the adjacent subcompartments. Unidirectional arrow is the active transport on the cell membrane. Metabolism can occur in any subcompartment.

**Figure 3 ijms-24-16224-f003:**
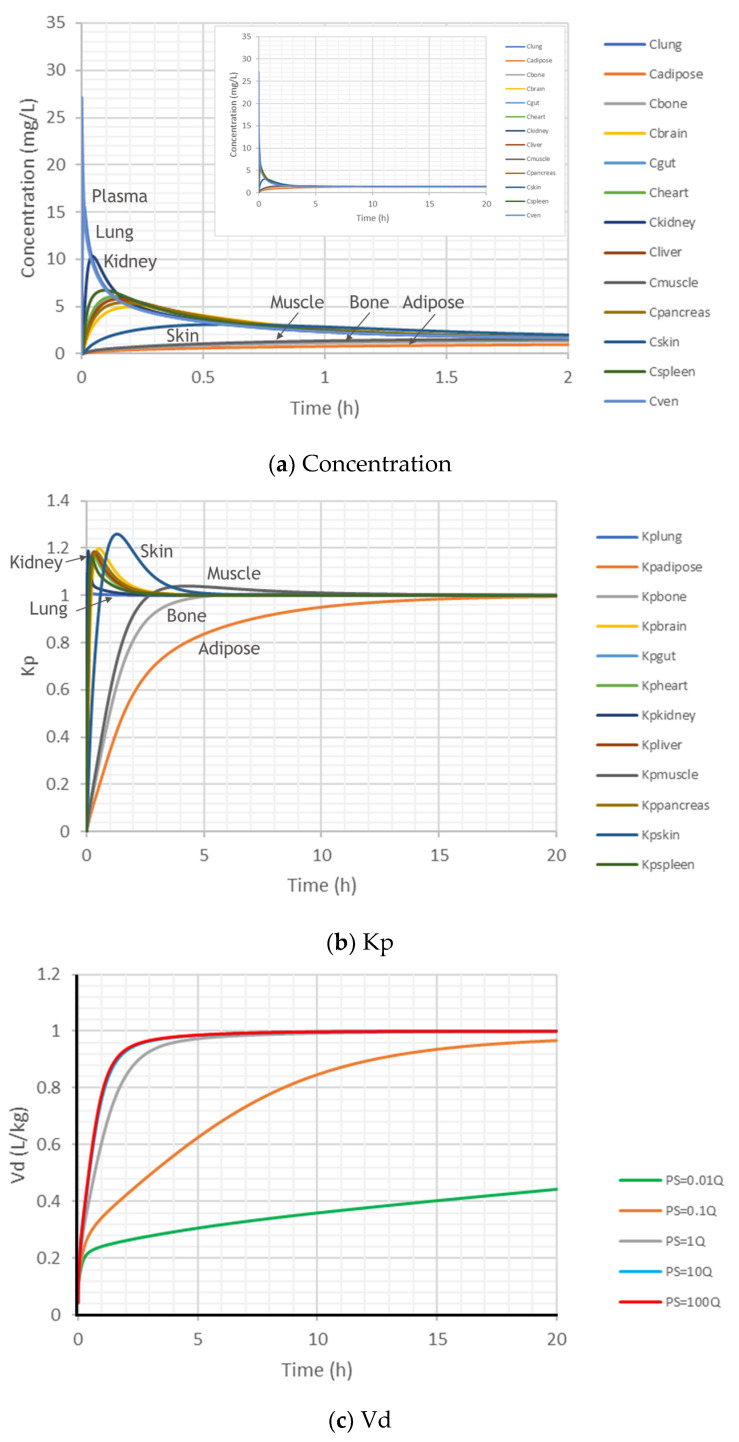
Simulated time profiles of tissue concentrations (**a**), Kp (**b**), and Vd (**c**) of a closed system with no active transport or metabolism after IV bolus 100 mg. (**a**) shows the first 2 h simulated concentration profiles of the 12 tissues and the venous plasma after an IV bolus. Insert in (**a**) is the whole concentration–time profiles over the simulated 20 h. (**b**) shows the simulated Kp profiles in the 12 tissues. (**c**) shows the simulated Vd profiles when the passive permeability (PS) was changed as 0.01- to 100-fold of tissue blood flow (Q). Note that the Vd profiles for PS = 10Q and PS = 100Q are overlapped in (**c**).

**Figure 4 ijms-24-16224-f004:**
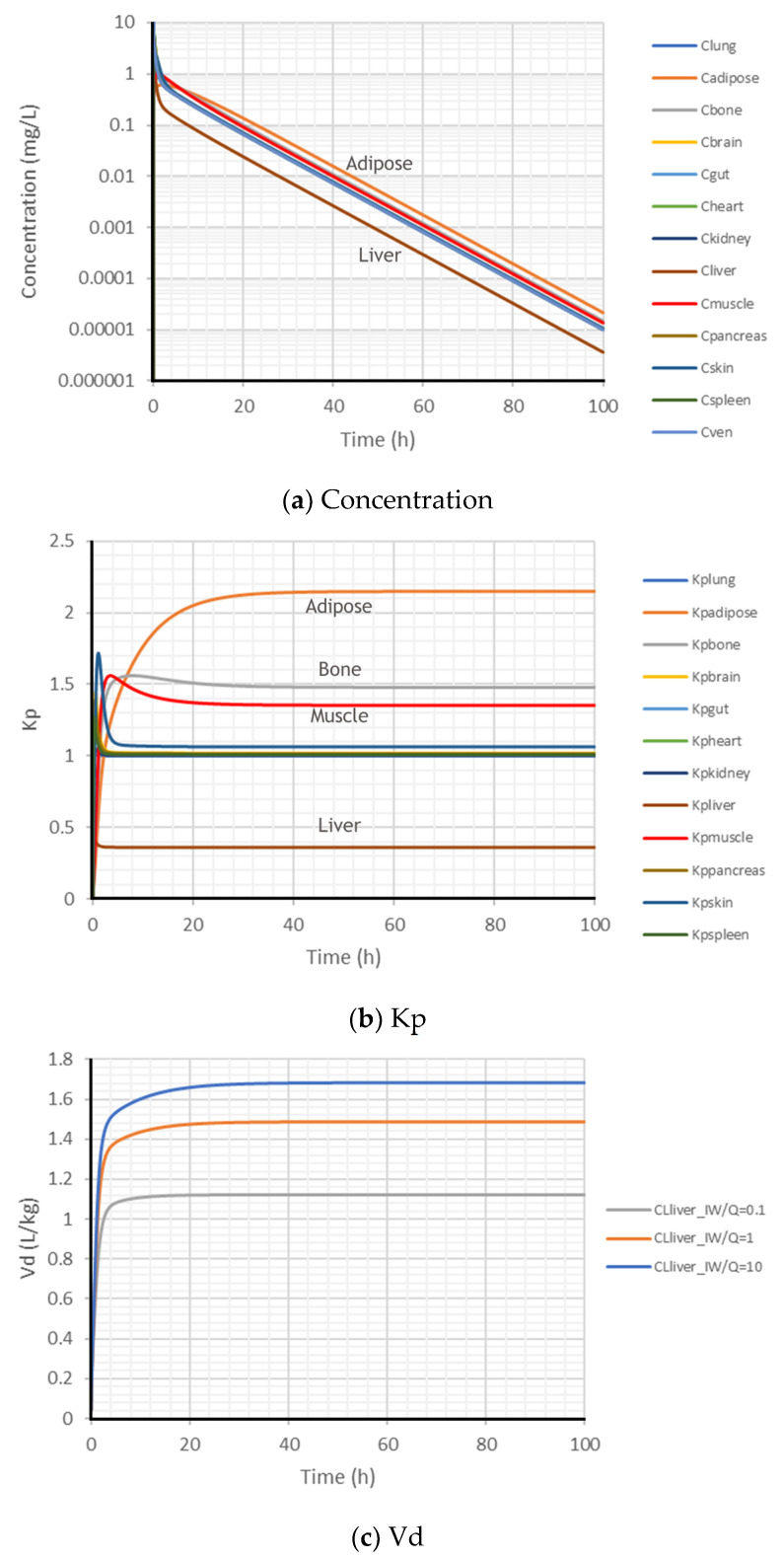
Simulated time profiles of tissue concentrations (**a**), Kp (**b**), and Vd (**c**) of an open system with hepatic metabolism after IV bolus 100 mg. (**a**) shows the simulated concentration profiles of the 12 tissues and the venous plasma after an IV bolus when the intrinsic clearance in the liver intracellular water (CLliver_IW) was the same as hepatic blood flow (Q). (**b**) shows the simulated Kp profiles in the 12 tissues. (**c**) shows the simulated Vd profiles when CLliver_IW was changing to 0.1- to 10-fold of hepatic blood flow (Q).

**Figure 5 ijms-24-16224-f005:**
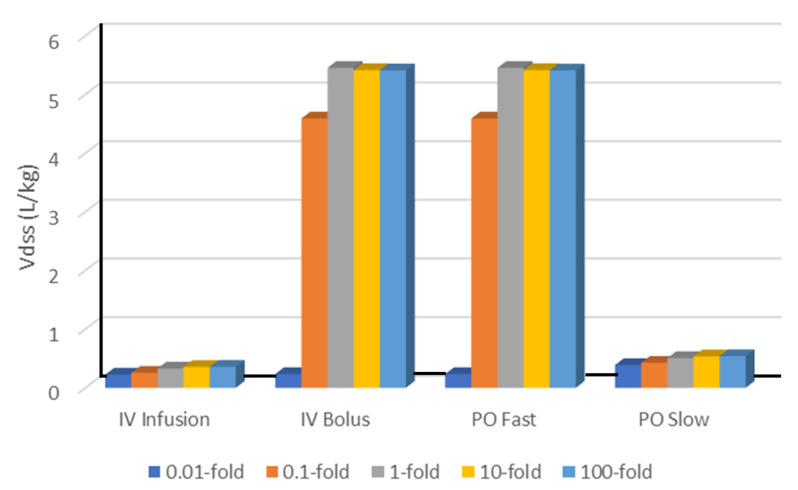
Impact of passive permeability on Vdss with intracellular metabolism (CL_IW/Q = 1) in all tissues. Fold relationships are passive permeability (PS) relative to tissue blood flow (Q).

**Figure 6 ijms-24-16224-f006:**
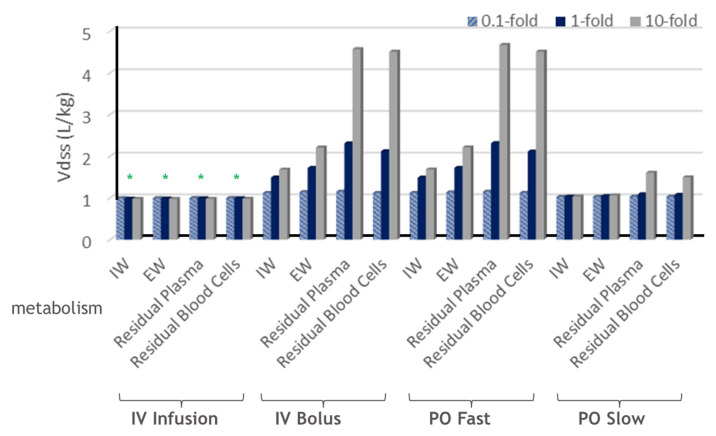
Summary of Vdss with respect to metabolism in liver. * indicates decreased Vdss with respect to increasing hepatic metabolism. Fold relationships are hepatic metabolism (CL) relative to hepatic blood flow (Q).

**Figure 7 ijms-24-16224-f007:**
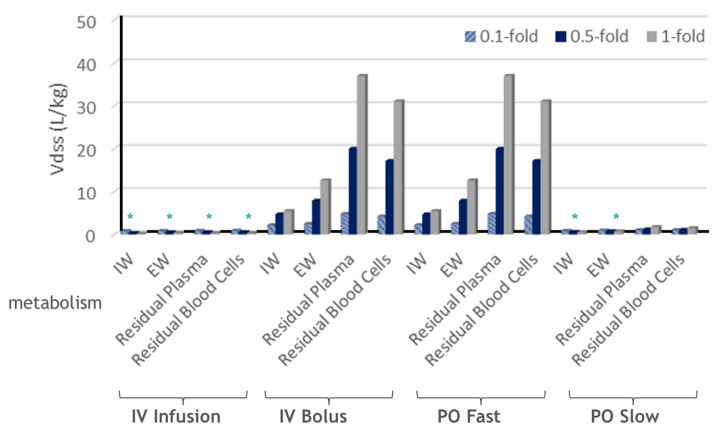
Summary of Vdss with respect to metabolism in all tissues. * indicates decreased Vdss with respect to increasing tissue metabolism. Fold relationships are metabolism in all tissues (CL) relative to blood flow (Q).

**Figure 8 ijms-24-16224-f008:**
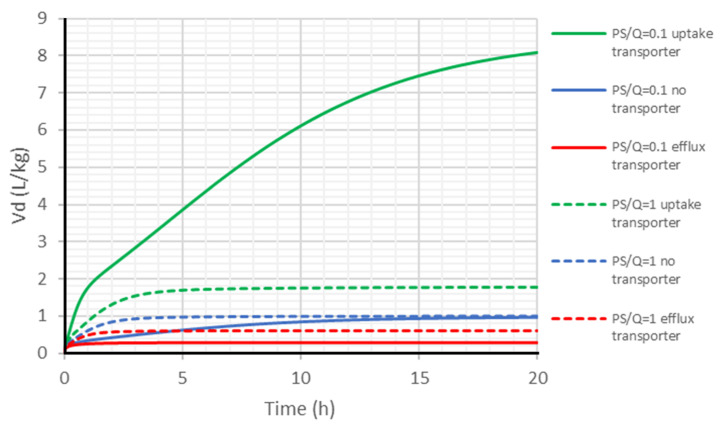
Impact of transporters on Vd for a closed system with no metabolism after IV bolus 100 mg. Passive permeability (PS) was 0.1-fold or the same as tissue blood flow. When activated, uptake transporter or efflux transporter clearance was set the same as tissue blood flow in all tissues.

**Figure 9 ijms-24-16224-f009:**
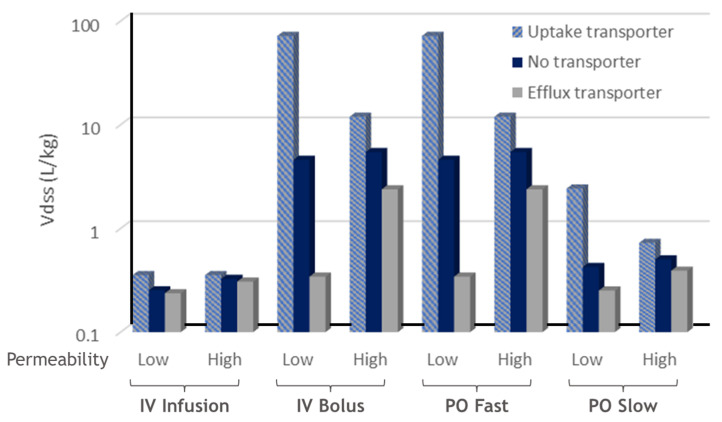
Summary of Vdss with respect to transporters in all tissues.

**Figure 10 ijms-24-16224-f010:**
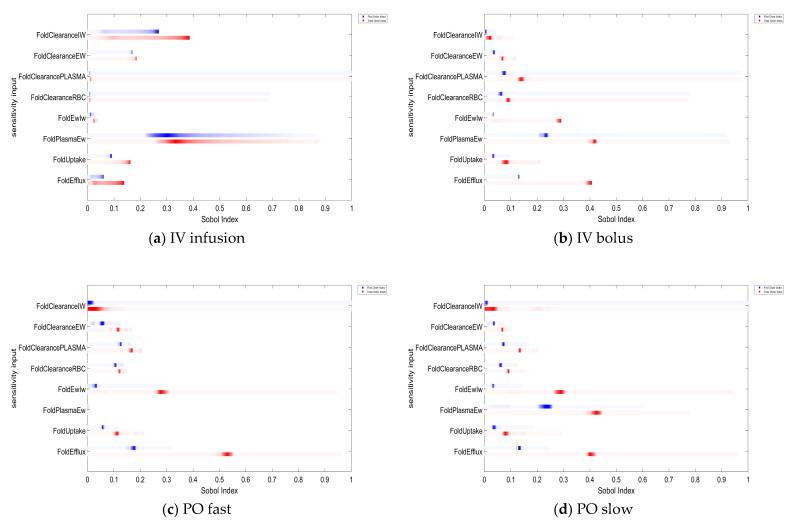
Sobol index for subcompartment metabolism, passive permeability, and active transporters. FoldClearanceIW, FoldClearanceEW, FoldClearancePLASMA, and FoldClearanceRBC are ratios to define the metabolic clearances in the IW, EW, TP, and TC subcompartments, respectively. FoldEwIw and FoldPlasmaEw are ratios to define the passive permeability between EW and IW subcompartments and that between TP and EW subcompartments. FoldUptake and FoldEfflux are ratios to define the uptake transporter and efflux transporter on the cell membrane which is between the IW and EW subcompartments. All ratios were based on the tissue blood flow in each tissue. The blue bar and red bar are the first-order index and the total order Sobol index. Darker colors mean that those values occur more often over the whole time course.

**Table 1 ijms-24-16224-t001:** Impact of permeability on Vdss with metabolism in all tissues.

Dosing	Vdss (L/kg)				
	PS/Q 0.01-Fold	PS/Q 0.1-Fold	PS/Q 1-Fold	PS/Q 10-Fold	PS/Q 100-Fold
IV Infusion	0.22	0.25	0.33	0.35	0.36
IV Bolus	0.23	4.59	5.45	5.41	5.41
PO Fast	0.23	4.59	5.45	5.41	5.41
PO Slow	0.39	0.43	0.50	0.53	0.54

PS/Q defines the fold of passive permeability over the tissue blood flow. CL_IW = Q for metabolic clearance in the intracellular water of all tissues.

**Table 2 ijms-24-16224-t002:** Summary of Vdss with respect to metabolism in liver.

Dosing	Metabolism in Liver Subcompartments		Vdss (L/kg)	
		CL/Qliver = 0.1	CL/Qliver = 1	CL/Qliver = 10
IV Infusion	IW	0.99	0.98	0.98
	EW	1.00	0.99	0.98
	Residual Plasma	1.00	0.99	0.98
	Residual Blood Cells	1.00	0.99	0.98
IV Bolus	IW	1.12	1.49	1.68
	EW	1.14	1.72	2.21
	Residual Plasma	1.15	2.31	4.57
	Residual Blood Cells	1.12	2.12	4.51
PO Fast	IW	1.12	1.49	1.68
	EW	1.14	1.72	2.21
	Residual Plasma	1.15	2.32	4.67
	Residual Blood Cells	1.12	2.11	4.51
PO Slow	IW	1.03	1.03	1.04
	EW	1.03	1.05	1.06
	Residual Plasma	1.03	1.09	1.61
	Residual Blood Cells	1.03	1.08	1.50

CL/Qliver defines the fold of metabolism over the hepatic blood flow.

**Table 3 ijms-24-16224-t003:** Summary of Vdss with respect to metabolism in all tissues.

Dosing	Metabolism in Tissue Subcompartments		Vdss (L/kg)	
		CL/Q = 0.1	CL/Q = 0.5	CL/Q = 1
IV Infusion	IW	0.79	0.33	0.15
	EW	0.83	0.51	0.36
	Residual Plasma	0.85	0.50	0.30
	Residual Blood Cells	0.87	0.54	0.34
IV Bolus	IW	2.17	4.69	5.45
	EW	2.47	7.86	12.59
	Residual Plasma	4.79	19.97	36.97
	Residual Blood Cells	4.19	17.13	31.05
PO Fast	IW	2.17	4.69	5.45
	EW	2.47	7.86	12.59
	Residual Plasma	4.79	19.97	36.97
	Residual Blood Cells	4.19	17.13	31.05
PO Slow	IW	0.86	0.60	0.50
	EW	0.91	0.74	0.73
	Residual Plasma	1.00	1.16	1.71
	Residual Blood Cells	1.00	1.09	1.46

CL/Q defines the fold of metabolism over the tissue blood flow.

**Table 4 ijms-24-16224-t004:** Summary of Vdss with respect to transporters in all tissues.

Dosing	Vdss (L/kg)			
	Permeability	Uptake Transporter	No Transporter	Efflux Transporter
IV Infusion	Low	0.36	0.25	0.24
	High	0.36	0.33	0.31
IV Bolus	Low	71.90	4.59	0.34
	High	11.96	5.45	2.38
PO Fast	Low	71.90	4.59	0.34
	High	11.96	5.45	2.38
PO Slow	Low	2.43	0.43	0.25
	High	0.73	0.50	0.39

## Data Availability

Data are contained within the article or Appendix A.

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
