# Peer review of "The Interplay of Permeability, Metabolism, Transporters, and Dosing in Determining the Dynamics of the Tissue/Plasma Partition Coefficient and Volume of Distribution—A Theoretical Investigation Using Permeability-Limited, Physiologically Based Pharmacokinetic Modeling"

_ijms, 2023, doi:10.3390/ijms242216224_

Round 1

Reviewer 1 Report

Comments and Suggestions for Authors

The manuscript entitled “Interplay of permeability, metabolism, transporters, and dosing 2 in determining the dynamics of tissue/plasma partition coefficient (Kp) and volume of distribution (Vd) – a theoretical investigation using permeability-limited physiologically-based 5 pharmacokinetic (PBPK) modeling” has been written with a focus on the effect of various drug absorption and elimination criteria using PBPK modeling approach. However, the manuscript needs further corrections to be accepted for the journal, which is as follows:

Line no. 140,161, 168, 172, 176,197, 202, 211, 219,253, 273,285, 317, 462: reference detail is not captured. Please add proper references.

Line no. 77: need to rephrase with grammatical correction.

Line no. 84: Please explain the abbreviations.

Line no. 108: needs to be re-phrase.

Line no. 130: Please provide details in the supplementary file.

Figure no. 1, 2, 3, 4,5, 6, 7 Y-axis and x-axis lines and titles need to be darkened in color.

Lines no. 545, and 564, increase the front side of equations.

Line no. 778: The figure number and caption color need to be changed.

Line no.  779 &780: repetition of table title.

 The manuscript has so many discrepancies, which need to be fulfilled and require further evaluation after correction of the above flaws.

Comments on the Quality of English Language

Rephrase the complex sentences to make it understandable to the PK Community.

Author Response

Dear Reviewer,

Thank you very much for reviewing and commenting on our manuscript. We followed your suggestion and have considered your comments point-by-point in our revision. Please find the attached our responses in the attachment. Hope our revision is acceptable.

Best Regards,

Lu Gaohua on behalf of the co-authors

Reviewer 2 Report

Comments and Suggestions for Authors

This is a great piece of research.

-in introduction, It is recommended that authors reason why "In fact, in a simulation 95 using Simcyp perfusion-limited PBPK model with hepatic metabolism alone, the tis- 96 sue/plasma concentration ratio (Kpss) at pseudo-steady-state in non-metabolic tissue is 97 higher than the constant Kptissue used in the above eq. (1)"

- In the results section the links to figures are shown as errors. The links in manuscript should be removed before saving it as pdf

- Could authors comment about the impact of physchecm properties like logp and fu on kps in open system? in the example with bolus and hepatic metabolism kp<1 is expected but how do you explain adipose kp? basically why kp goes over 1 in non-metabolic tissue?

- I understand that it is extremely rare to have these kind of data in preclinical species, however, it would have been nice to have a proof of concept

Comments on the Quality of English Language

English is fine.

Author Response

(The authors gave the same response as above.)
